# Nanodomain coupling explains Ca²⁺ independence of transmitter release time course at a fast central synapse

Itaru Arai, Peter Jonas*

IST Austria (Institute of Science and Technology Austria), Klosterneuburg, Austria

**Abstract** A puzzling property of synaptic transmission, originally established at the neuromuscular junction, is that the time course of transmitter release is independent of the extracellular Ca²⁺ concentration ([Ca²⁺]$_o$), whereas the rate of release is highly [Ca²⁺]$_o$-dependent. Here, we examine the time course of release at inhibitory basket cell-Purkinje cell synapses and show that it is independent of [Ca²⁺]$_o$. Modeling of Ca²⁺-dependent transmitter release suggests that the invariant time course of release critically depends on tight coupling between Ca²⁺ channels and release sensors. Experiments with exogenous Ca²⁺ chelators reveal that channel-sensor coupling at basket cell-Purkinje cell synapses is very tight, with a mean distance of 10–20 nm. Thus, tight channel-sensor coupling provides a mechanistic explanation for the apparent [Ca²⁺]$_o$ independence of the time course of release.

## Introduction

Calcium plays a key role in the control of transmitter release at chemical synapses (*Neher, 1998*; *Südhof, 2013*). As transmitter release is a Ca²⁺-dependent biochemical process, one would expect that both the extent and kinetics of release will depend on Ca²⁺ concentration. In contrast to this expectation, the time course of release (TCR) at the neuromuscular junction is independent of the extracellular Ca²⁺ concentration ([Ca²⁺]$_o$) (*Datyner and Gage, 1980*; *Van der Kloot, 1988*; *Parnas et al., 1989*). However, the apparent [Ca²⁺]$_o$ independence of the TCR is less well established at central synapses (*Sargent et al., 2005*). Furthermore, the underlying mechanisms remain elusive. Several potential explanations were proposed, including adaptation of transmitter release (*Hsu et al., 1996*) and additional voltage sensors that control the timing of release (*Parnas et al., 2000*; See; *Felmy et al., 2003*). Recent results suggested that in several central synapses presynaptic Ca²⁺ channels and release sensors are tightly coupled (*Bucurenciu et al., 2008*; *Eggermann et al., 2012*; *Scimemi and Diamond, 2012*; *Schmidt et al., 2013*). Tight coupling might be another factor contributing to the apparent [Ca²⁺]$_o$ independence of the TCR (*Yamada and Zucker, 1992*). However, this possibility has not been directly examined.

## Results

We studied the [Ca²⁺]$_o$ dependence of transmitter release at GABAergic synapses between cerebellar basket cells (BCs) and Purkinje cells (PCs), using paired whole-cell recordings from synaptically connected neurons in mouse brain slices (*Caillard et al., 2000*; *Sakaba, 2008*; *Eggermann and Jonas, 2012*; *Figure 1A–D*). Action potentials were evoked in the presynaptic BC in whole-cell current-clamp or cell-attached voltage-clamp configurations, whereas IPSCs were recorded in the postsynaptic PC under whole-cell voltage-clamp conditions (series resistance 3.8 ± 0.1 MΩ; mean ± SEM; 92 cells; 'Materials and methods'). Unitary IPSCs were initiated with short latency and high temporal precision. Synaptic latency was 1.20 ± 0.03 ms at ~22°C (24 pairs) and 0.47 ± 0.02 ms at ~34°C (5 pairs; *Figure 1C,D*).

*For correspondence: peter. jonas@ist.ac.at

**Competing interests:** The authors declare that no competing interests exist.

**Reviewing editor**: Michael Häusser, University College London, United Kingdom

**eLife digest** The nervous system sends information around the body in the form of electrical signals that travel through cells called neurons. However, these electrical signals cannot cross the synapses between neurons. Instead, the information is carried across the synapse by molecules called neurotransmitters.

Calcium ions control the release of neurotransmitters. There is a high concentration of calcium ions outside the neuron but they are not able to pass through the cell membrane under normal conditions. However, when an electrical impulse reaches the synapse, ion channels in the membrane open and allow calcium ions to enter the cell. Once inside, the ions activate the release of neurotransmitters by binding to proteins called release sensors.

Several experiments on the release of neurotransmitters have studied the synapses between neurons and muscle fibers. These studies found that the higher the concentration of ions outside the neuron, the higher the rate at which the neurotransmitters were released. However, the timing of release—the length of time over which the neurotransmitters were released—did not depend on the concentration of calcium ions.

Arai and Jonas have now studied neurotransmitter release at a synapse in a region of the brain called the cerebellum. These experiments also found that the timing of the release did not depend on the ion concentration, suggesting that this may be a general property of neurotransmitter release.

To find out more, Arai and Jonas created a mathematical model of neurotransmitter release. This model suggests that for the timing of release to remain the same, the ion channel and the release sensor must be located close together in the presynaptic terminal. If they are not close together, the timing of release becomes blurred and more dependent on the external calcium concentration.

Further experiments confirm the prediction of the model by showing that the calcium channels and the release sensors in these synapses are very close together. The next challenge will be to find out whether the conclusions are also valid for other synapses where the calcium channels and release sensors are further apart.

Synaptic transmission was entirely blocked by bath application of the selective P/Q-type $Ca^{2+}$ channel blocker ω-agatoxin IVa (1 μM), whereas the N-type $Ca^{2+}$ channel blocker ω-conotoxin GVIa (1 μM) had no detectable effect (*Figure 1E–H*). Thus, transmitter release at this synapse is selectively mediated by P/Q-type $Ca^{2+}$ channels.

Exploiting the technical advantages of this synapse (including ideal voltage-clamp conditions conveyed by perisomatic synapse location, presynaptic accessibility due to short axon trajectories, and optimal signal-to-noise ratio because of large quantal size), we first examined the $[Ca^{2+}]_o$ dependence of release efficacy. Analysis of the dependence of evoked IPSC peak amplitude on $[Ca^{2+}]_o$ revealed that the amount of transmitter release was steeply $[Ca^{2+}]_o$-dependent, with a high power coefficient (n = 3.02 in the low-concentration limit; *Figure 2A,B*). Thus, release at the BC–PC synapse was highly cooperative (*Dodge and Rahamimoff, 1967*). We next measured the $[Ca^{2+}]_o$ dependence of release kinetics. To determine the TCR, we first recorded unitary IPSCs at a given $[Ca^{2+}]_o$, and subsequently in reduced $[Ca^{2+}]_o$ to isolate quantal IPSCs (*Figure 2C*). We then computed the TCR by deconvolution of unitary and quantal IPSC waveforms (*Diamond and Jahr, 1995*; *Neher and Sakaba, 2001*; *Sakaba, 2008*; *Figure 2D*). When $[Ca^{2+}]_o$ was changed in the range from 0.7–4 mM, peak release rate changed by a factor of 34.4. In contrast, the half-duration of the TCR was only minimally affected (0.49 ± 0.05 ms at 0.7 mM; 0.43 ± 0.05 ms at 1 mM; 0.47 ± 0.01 ms at 2 mM; 0.47 ± 0.02 ms at 4 mM; 5–11 pairs; p = 0.56; *Figure 2D,E*). Furthermore, the decay time constant of the TCR was similar in the different conditions (p = 0.38; *Figure 2D,E*). These results show that the TCR is largely $[Ca^{2+}]_o$-independent at a central synapse. Similar conclusions were reached at lower recording temperature, which would be expected to markedly slow down the kinetic rates of channels and sensors, but to only minimally change the rates of diffusional processes. At ~12°C, changes in $[Ca^{2+}]_o$ had no detectable effects on the half-duration of the TCR (2.84 ± 0.42 ms at 1 mM, 2.97 ± 0.24 ms at 2 mM, and 2.79 ± 0.25 ms at 4 mM $[Ca^{2+}]_o$; 5 pairs; p = 0.93) (*Figure 2—figure supplement 1*).

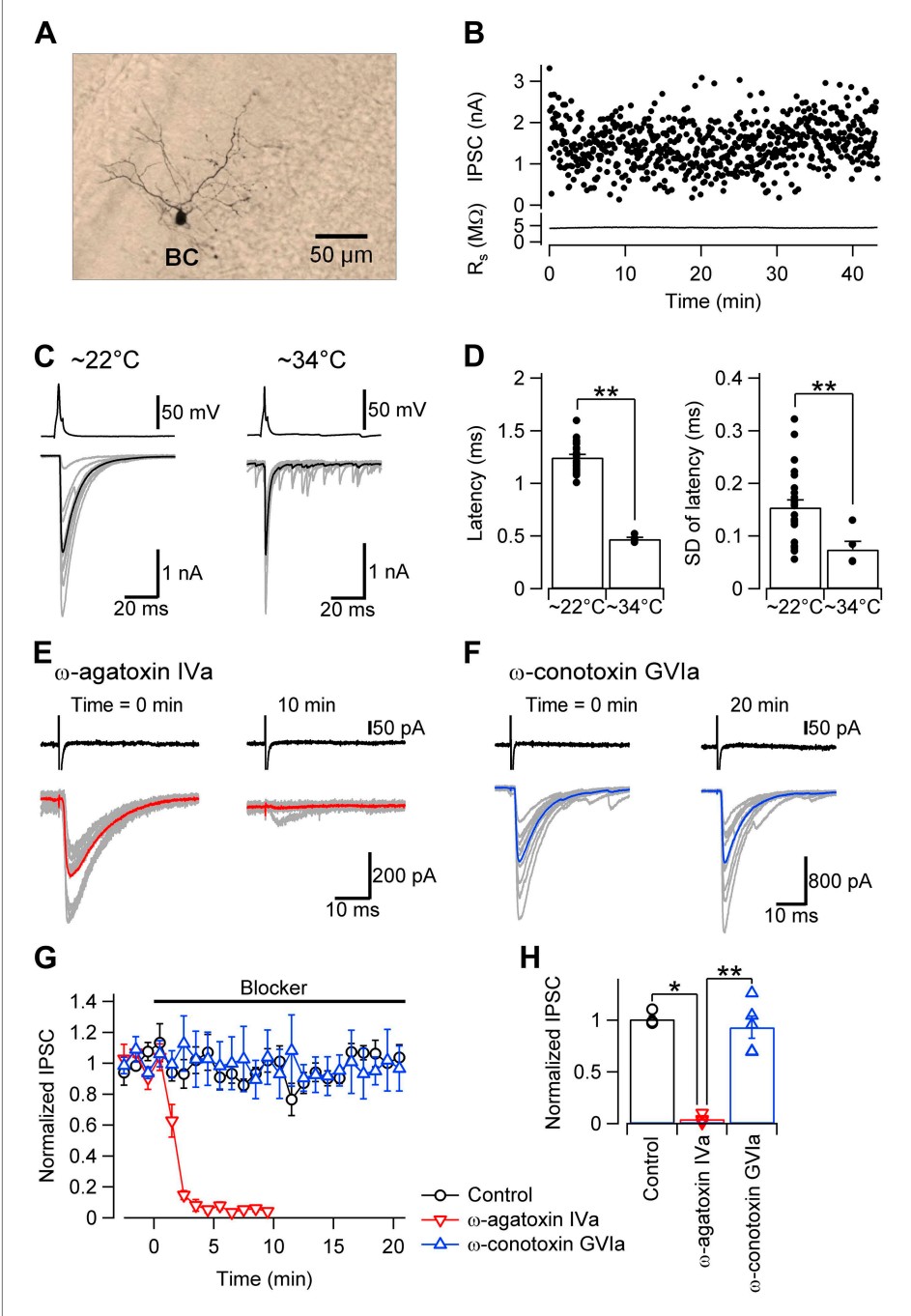

**Figure 1**. Fast and synchronous transmitter release at BC–PC synapses in the cerebellum is exclusively mediated by P/Q-type $Ca^{2+}$ channels. (**A**) Light micrograph of a cerebellar basket cell filled with biocytin during recording and labeled using 3,3'-diaminobenzidine as chromogen. Similar morphological properties were obtained in 29 other biocytin-labeled cerebellar BCs. (**B**) Plot of IPSC peak amplitude (top) and corresponding series resistance (bottom) against time during recording. Action potentials were evoked in the presynaptic cell at time intervals of 4 s. Note that evoked IPSC shows only little rundown for more than 40 min with stable series resistance. (**C**) Presynaptic action potentials evoked in the whole-cell current clamp configuration (top) and evoked IPSCs (bottom) recorded at ~22°C (left) and ~34°C (right). 10 consecutive individual traces (gray) and the corresponding average trace (black) are shown superimposed. (**D**) Summary bar graph. Left, latency (time between steepest point in the rising phase of the presynaptic action potential and IPSC onset) at ~22°C and ~34°C. Right, standard deviation of latency, a measure of synchrony of transmitter release. Bars indicate mean ± SEM; solid circles represent data from individual

*Figure 1. Continued on next page*

*Figure 1. Continued*

experiments (~22°C: 24 pairs; ~34°C: 5 pairs). (**E**) IPSCs in a BC–PC pair before (left) and after (right) application of 1 µM of the selective P/Q-type $Ca^{2+}$ channel blocker ω-agatoxin IVa. Synaptic transmission was almost completely blocked. Top, presynaptic action currents evoked in the cell-attached voltage-clamp configuration; bottom, corresponding IPSCs. 10 consecutive individual traces (gray) and the corresponding average trace (red) are shown superimposed. (**F**) Similar recording as shown in (**E**), but with 1 µM of the selective N-type $Ca^{2+}$ channel blocker ω-conotoxin GVIa. (**G**) Plot of IPSC peak amplitude against time during application of 1 µM of ω-agatoxin IVa (red) or ω-conotoxin GVIa (blue). The time of application of the different extracellular solutions is represented by horizontal bars. Black: mock application of control solution. Symbols indicate mean; error bars represent SEM. (**H**) Summary bar graph of the effects of $Ca^{2+}$ channel blockers. Bars indicate mean ± SEM; solid circles represent data from individual experiments (control: 4 pairs; ω-agatoxin IVa: 4 pairs; ω-conotoxin GVIa: 5 pairs). To achieve maximal stability, presynaptic BCs were noninvasively stimulated in the cell-attached configuration in all experiments. * and ** indicate p < 0.05 and 0.01, respectively. All experiments except subsets in (**C**), (**D**) were performed at ~22°C.

To explore the mechanisms underlying the paradoxical $[Ca^{2+}]_o$ independence of the TCR, we used a realistic model of action potential-dependent transmitter release (*Eggermann and Jonas, 2012*; *Figure 3*). $Ca^{2+}$ diffusion and buffering were computed by solving the full set of partial differential diffusion and reaction equations, and the TCR was simulated using a previously established release sensor model (*Lou et al., 2005*; *Figure 3A,B*). Finally, the half-duration of the TCR was plotted against the peak release rate (PRR). Computational analysis revealed that the TCR was markedly dependent on the PRR in both low and high PRR limit. In the low PRR limit, the half-duration of the TCR decreased with PRR in a model with fixed coupling distance (*Figure 3C*). This dependence was accentuated after slowing of release sensor rates (*Figure 3D*, top), indicating that rate-limiting sensor kinetics are responsible. Furthermore, this dependence was inverted in a model with variable coupling distance (*Figure 3E*). In such a configuration, small $Ca^{2+}$ inflow may selectively release proximal vesicles, whereas large $Ca^{2+}$ inflow may recruit all vesicles, resulting in a broadening of the TCR. In the high PRR limit, the half-duration of the TCR increased with PRR (*Figure 3C*). This behavior was particularly prominent in small boutons (*Figure 3F*, bottom), suggesting that saturation of the endogenous buffers leads to broadening of the TCR. In all configurations, the dependence of the TCR on the PRR was substantially more prominent in loose (200 nm, blue colors) than in tight coupling configurations (20 nm, red colors). Similar conclusions were reached when the concentration and unbinding rate of fixed buffers were altered (*Figure 3—figure supplement 1*; *Gilmanov et al., 2008*). Thus, whereas several factors influence the shape of the TCR–PRR relation, the coupling distance plays a key role in all conditions examined.

Our modeling results suggest the intriguing possibility that tight coupling between $Ca^{2+}$ channels and release sensors (*Christie et al., 2011*) might be a main reason for the apparent $[Ca^{2+}]_o$ independence of the TCR at the cerebellar BC–PC synapse. To test this hypothesis quantitatively, we probed the distance between $Ca^{2+}$ channels and release sensors, using exogenous $Ca^{2+}$ chelators (*Adler et al., 1991*; *Neher, 1998*; *Eggermann et al., 2012*; *Figure 4*). $Ca^{2+}$ chelators were loaded into presynaptic BC terminals through the somatic patch pipette, and the reversibility of their effects was tested by obtaining a second recording from the same presynaptic BC at later times. Whereas the fast $Ca^{2+}$ chelator ethylenedioxybis-(o-phenylenenitrilo)-N,N,N',N'-tetraacetic acid (BAPTA) markedly suppressed transmitter release (*Figure 4A,B*), the slow $Ca^{2+}$ chelator ethyleneglycol-bis(2-aminoethylether)-N,N,N',N'-tetraacetic acid (EGTA) was much less effective (*Figure 4C,D*). Analysis of concentration–effect data revealed that the half-maximal inhibitory concentration ($IC_{50}$) for the steady-state effect was 0.6 mM for BAPTA (14 pairs total) and 16.0 mM for EGTA (11 pairs total; *Figure 4E*). Thus, the $IC_{50}$ value was ~27-fold higher for EGTA than for BAPTA.

To estimate the coupling distance, we fit the concentration–effect data for BAPTA and EGTA with a model of $Ca^{2+}$ diffusion and buffering based on linear approximations (*Neher, 1998*; *Figure 4F*). The main free parameter in the model was the distance between $Ca^{2+}$ channels and release sensors, while several other parameters (e.g. the physicochemical properties of the $Ca^{2+}$ chelators and the cooperativity of transmitter release) were well constrained. Analysis of the entire data set revealed a coupling distance of 10–20 nm (11.4 nm for constant coupling distance; 13.9 nm for half-normally distributed coupling distance; 10.1 nm for skewed-normally distributed coupling distance; *Figure 4F*). Statistical errors, as assessed by bootstrap analysis, were minimal (1.2 nm; *Figure 4G*). Furthermore, systematic

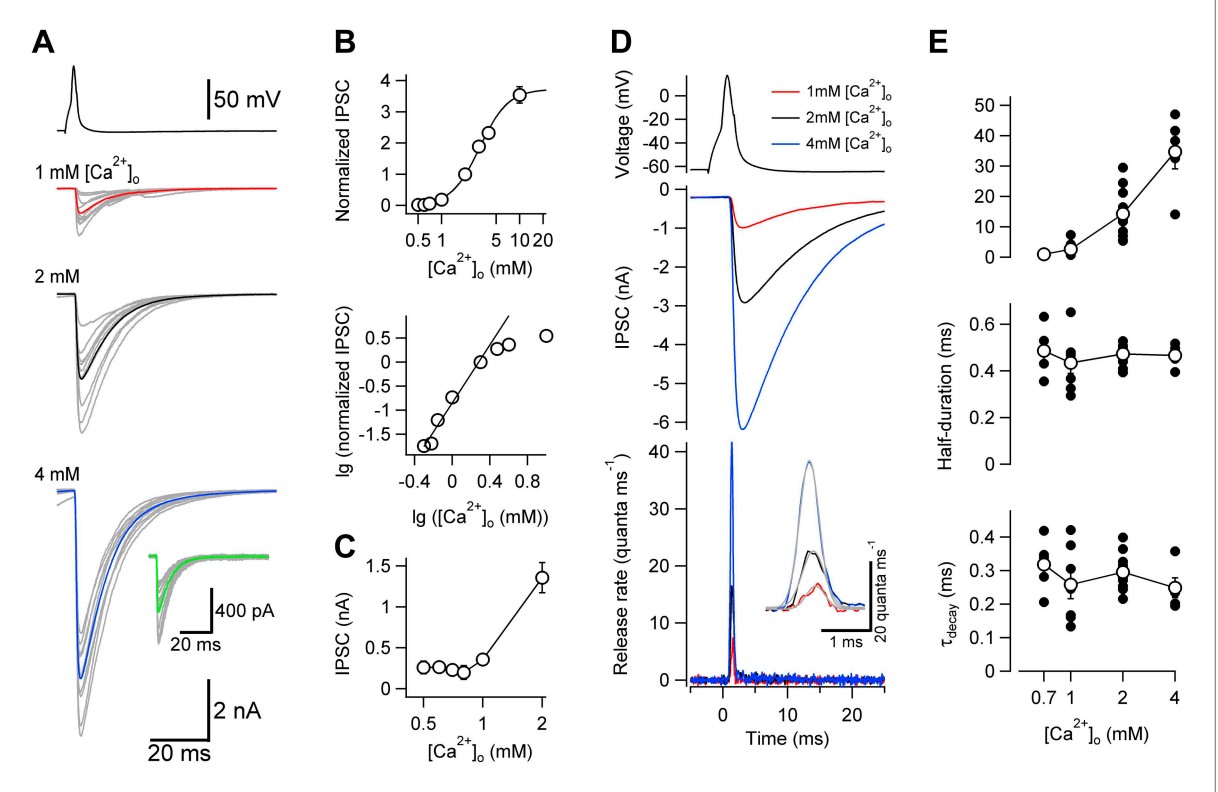

**Figure 2**. Changes in [Ca²⁺]ₒ affect peak release rate, but leave the TCR unaffected at the cerebellar BC–PC synapse. (**A**) Presynaptic action potential (top) and evoked IPSCs (bottom) in different [Ca²⁺]ₒ. 10 consecutive individual traces (gray) and the corresponding average trace (red, 1 mM [Ca²⁺]ₒ; black, 2 mM; blue, 4 mM) are shown superimposed. Inset shows quantal IPSCs; 10 individual IPSCs recorded in 0.7 mM [Ca²⁺]ₒ (gray) and the corresponding average trace (green) are shown superimposed. (**B**) Relationship between IPSC peak amplitude and [Ca²⁺]ₒ. IPSC amplitudes were normalized to the value at 2 mM [Ca²⁺]ₒ and averaged across pairs. Top, linear–logarithmic representation. Data were fit with a Hill equation, yielding a maximal value (a) of 3.73, a half-maximal effective concentration (EC₅₀) of 3.09 mM, and an average Hill coefficient (n) of 2.39. Bottom, double-logarithmic plot; data points for [Ca²⁺]ₒ ≤ 2 mM were analyzed by linear regression, yielding a Hill coefficient of 3.02 in the low concentration limit. (**C**) Relationship between peak amplitude of IPSC successes and [Ca²⁺]ₒ. Symbols indicate mean, error bars represent SEM. The peak amplitude of the successes reaches a constant level below 1 mM [Ca²⁺]ₒ, suggesting that the level of quantal IPSCs was reached. (**D**) TCR in different [Ca²⁺]ₒ obtained by deconvolution. Top, presynaptic action potential. Middle, average evoked IPSCs in different [Ca²⁺]ₒ. Bottom, TCR in different [Ca²⁺]ₒ. Inset, expanded waveforms of the TCR (red, 1 mM [Ca²⁺]ₒ; black, 2 mM; blue, 4 mM; gray, fit Gaussian curves). Data in (**A**) and (**D**) are from the same pair. (**E**) Peak release rate (top), half-duration (middle), and decay time course of release period (bottom) plotted vs [Ca²⁺]ₒ. Open circles connected by lines indicate mean ± SEM; solid circles indicate data from individual experiments (0.7 mM: 5 pairs; 1 mM: 7 pairs; 2 mM: 11 pairs; 4 mM: 5 pairs). All experiments were performed at ~22°C. (Also see *Figure 2—figure supplement 1,2*).

The following figure supplements are available for figure 2:

**Figure supplement 1**. Lowering temperature slows TCR, but leaves its [Ca²⁺]ₒ independence unaffected.

**Figure supplement 2**. TCR is unlikely to be distorted by postsynaptic receptor saturation or desensitization.

errors were small, as indicated by the robustness of the estimate against variations in resting Ca²⁺ concentration and endogenous buffer product (*Figure 4H*). These results reveal that the coupling between Ca²⁺ channels and release sensors at the cerebellar BC–PC synapse is tight.

## Discussion

Our results address a long-standing question in the field of synaptic transmission (*Yamada and Zucker, 1992*; *Parnas et al., 2000*): If release probability is highly dependent on extracellular Ca²⁺ concentration (*Dodge and Rahamimoff, 1967*), how is it possible that the timing of release is [Ca²⁺]ₒ-independent (*Datyner and Gage, 1980*; *Van der Kloot, 1988*; *Parnas et al., 1989*)?

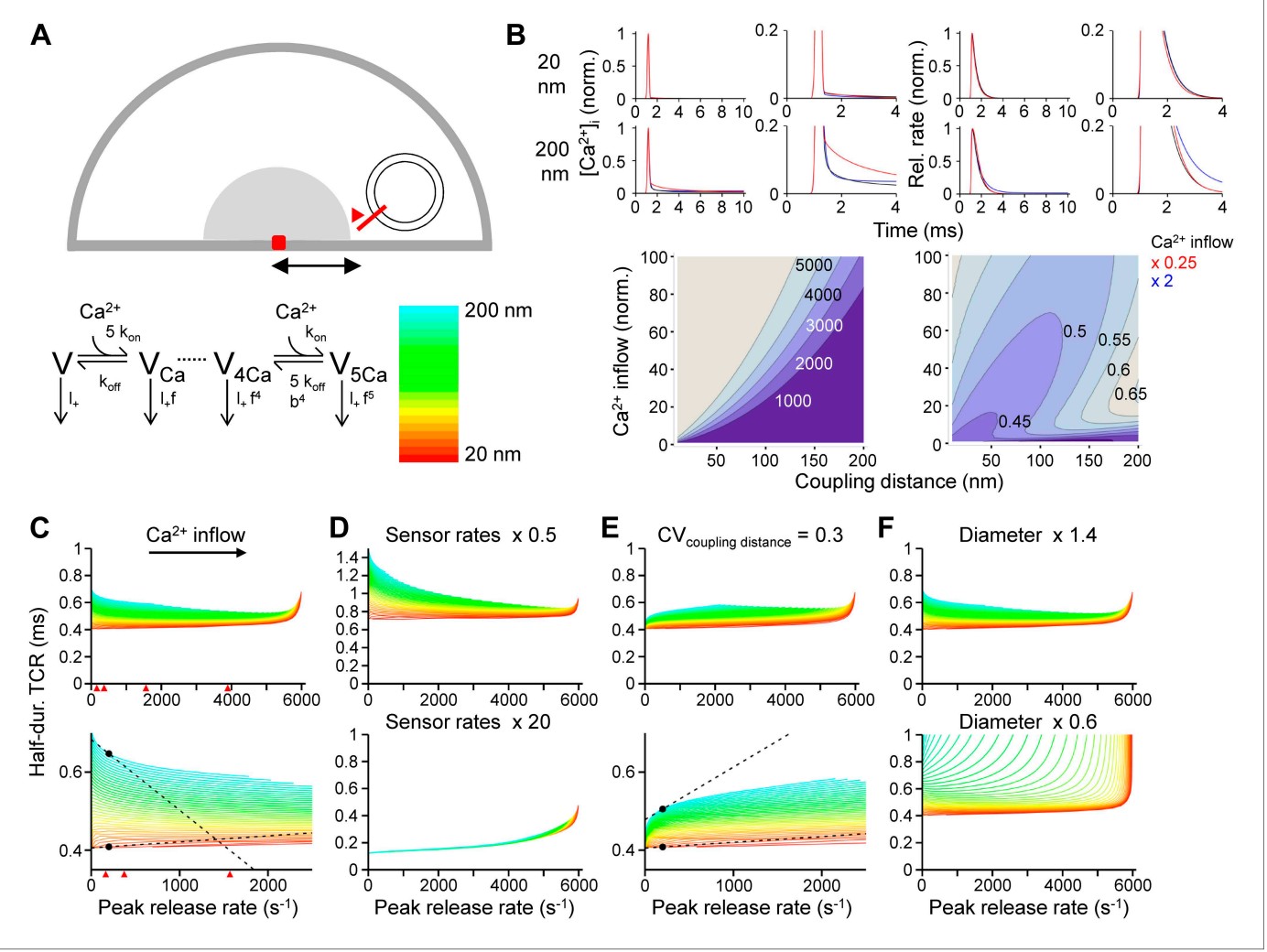

**Figure 3**. Tight Ca²⁺ channel–release sensor coupling reduces the $[Ca^{2+}]_o$ dependence of the TCR in a release model. (**A**) Schematic illustration of the release model. Top, location of Ca²⁺ source and release sensor within a schematic presynaptic terminal; bottom left, Ca²⁺ sensor model; bottom right, color code indicating coupling distance in subsequent plots (**C**–**F**). The presynaptic terminal was modeled as a hemisphere. Ca²⁺ inflow was generated by a point source in the center (red square and surrounding shaded area). The release sensor was placed at variable distance from the source (red line). Ca²⁺ transients were calculated as the numerical solution to the full set of partial differential reaction-diffusion equations (*Smith, 2001*). Transmitter release was computed using a modified version of a previously described sensor model (*Lou et al., 2005*). $k_{on}$ and $k_{off}$ are Ca²⁺-binding and unbinding rates, $l_+$ is basal release rate, and f and b are the cooperativity factors for release and Ca²⁺ unbinding, respectively. (**B**) Left, $[Ca^{2+}]_i$ plotted vs time. Right, corresponding release rate. Top, tight coupling (20 nm); bottom, loose coupling (200 nm distance). Black: default Ca²⁺ inflow (3.5 and 104.4 Ca²⁺ channel equivalents, leading to a vesicular release rate of ~2000 s⁻¹ in both cases); red: reduced Ca²⁺ inflow (x 0.25); blue: increased Ca²⁺ inflow (x 2). Bottom, contour plots of peak release rate (left) and half-duration of the TCR (right), plotted vs coupling distance (horizontal axis) and Ca²⁺ inflow (vertical axis, normalized to that of single Ca²⁺ channel). Numbers right-adjacent to the contour lines indicate peak release rate and half-duration of the TCR, respectively. Bouton diameter 1.0 µm. (**C**) Plot of half-duration of the TCR vs peak release rate for different coupling distances (individual curves for 10 to 200 nm in 5 nm steps; scale bar for color in (**A**)). Bottom graph shows expansion. Bouton diameter 1.0 µm. Red arrowheads indicate peak release rates at 0.7, 1, 2, and 4 mM $[Ca^{2+}]_o$, estimated from the IPSC–$[Ca^{2+}]_o$ curve in *Figure 2B*. (**D**) Half-duration of TCR–peak release rate relations for different sensor rates. Default sensor rates were slowed (x 0.5) or accelerated (x 20, to make the kinetics of the release sensor very fast in comparison to all other processes). (**E**) Half-duration of TCR–peak release rate relations for variable coupling distance (CV of 0.3). Bottom graph shows expansion. (**F**) Half-duration of TCR–peak release rate relations for different bouton diameters. Dashed lines and points in (**C**) and (**E**) indicate the slope of the TCR–peak release rate relations for 20 nm and 200 nm coupling distance at a release rate of 200 quanta s⁻¹. Note that the absolute value of the slope is ~12 times and ~7 times higher, respectively, for loose than for tight coupling. Also see *Figure 3—figure supplement 1*.

The following figure supplement is available for figure 3:

**Figure supplement 1**. Effects of concentration and affinity of the endogenous fixed buffer on the $[Ca^{2+}]_o$ dependence of the TCR in the release model.

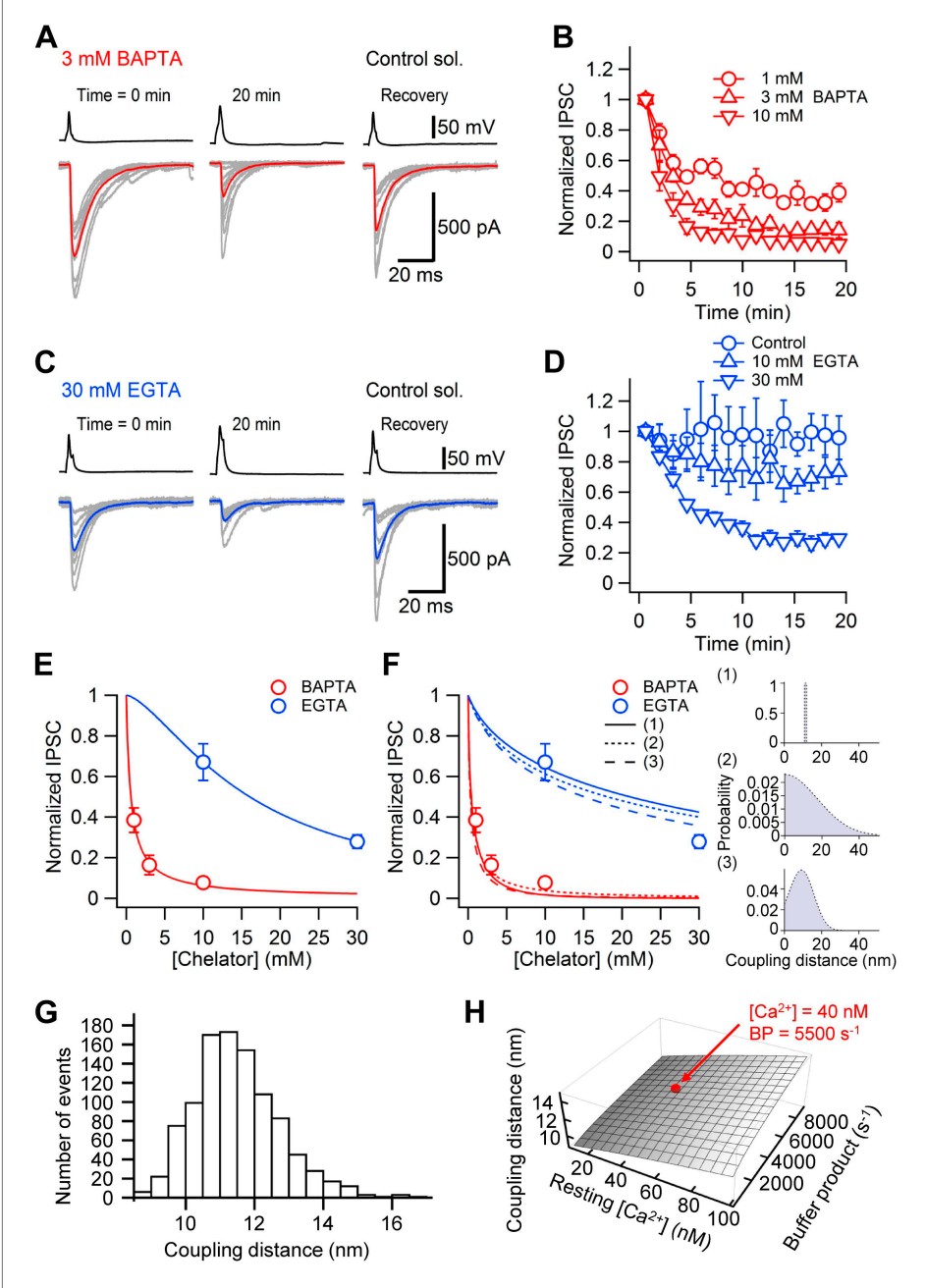

**Figure 4**. Nanodomain coupling between Ca²⁺ channels and release sensors at the cerebellar BC–PC synapse. (**A**) Presynaptic action potentials and evoked IPSCs recorded immediately after presynaptic BC break-in (left), 20 min after break-in to load 3 mM BAPTA into presynaptic terminals (middle), and after the presynaptic recoding pipette was changed to one containing control solution (right). 10 consecutive individual traces (gray) and the corresponding average trace (red) are shown superimposed. (**B**) Time course of the effect of 1 mM (○, 5 pairs), 3 mM (△, 4 pairs) and 10 mM (▽, 5 pairs) BAPTA. Data from each pair were normalized to the average amplitude obtained from the first 20 consecutive traces and averaged across pairs. (**C**) Similar recording as shown in (**A**), except that the internal solution for the presynaptic BC contained 30 mM EGTA. (**D**) Time course of the effect of 0.1 mM (○, control, 4 pairs), 10 mM (△, 5 pairs) and 30 mM (▽, 6 pairs) EGTA. In all experiments included in (**B**) and (**D**), the presynaptic BC was patched twice to demonstrate recovery. Symbols indicate mean, error bars represent SEM. (**E**) Plot of steady-state effects of chelators on IPSC peak amplitude against chelator concentration for BAPTA (red circles) and EGTA (blue circles). Curves represent Hill equations fit to the data points. IC₅₀ values were 0.6 mM and 16.0 mM, Hill coefficients were 0.96 and 1.51, respectively. (**F**) Same data, fit with linearized models of Ca²⁺ diffusion and

*Figure 4. Continued on next page*

*Figure 4. Continued*

buffering (continuous curve, constant coupling distance; small dashing, half-normally distributed coupling distance; large dashing, skewed-normally distributed coupling distance; corresponding distributions shown as insets on the right). For the single-channel model, the coupling distance was estimated as 11.4 nm. For the half-normal distribution, the standard deviation was 10.5 nm (expectation value 13.9 nm; skewness 1.0). For the skewed-normal distribution, location, scale, and shape parameter were 13.7 nm, 8.6 nm, and −1.11, respectively (expectation value 10.1 nm; skewness 0.40). (**G**) Statistical errors. Histogram of coupling distance estimates in 1000 bootstrap replications. (**H**) Systematic errors. Plot of coupling distance against resting $Ca^{2+}$ concentration and endogenous buffer product. Red point indicates default parameter values. In (**G**) and (**H**), the model with constant coupling distance was used. All experiments were performed at ~22°C.

To address this question, we measured the $[Ca^{2+}]_o$ dependence of both the amount and the time course of transmitter release at the cerebellar BC–PC synapse, a central synapse in which the TCR can be precisely quantified. Our results demonstrate that while the amount of release is highly $[Ca^{2+}]_o$-dependent with a power coefficient of ~3, the TCR is largely $[Ca^{2+}]_o$-independent. Although the apparent $[Ca^{2+}]_o$ independence of the TCR has been well established at the neuromuscular junction (*Datyner and Gage, 1980*; *Van der Kloot, 1988*; *Parnas et al., 1989*), this phenomenon is less well documented at central synapses (see *Sargent et al., 2005* for a notable exception). Thus, our results demonstrate that the $[Ca^{2+}]_o$ independence of the TCR is a general phenomenon characteristic for synaptic transmission in both peripheral and central nervous system.

We further measured the coupling distance between $Ca^{2+}$ channels and release sensors at BC–PC synapses. Our results show that coupling is tight, with a coupling distance of 10–20 nm. Although several previous studies measured the coupling distance, the rules that define the coupling configuration remain elusive (*Hefft and Jonas, 2005*; *Bucurenciu et al., 2008*; *Christie et al., 2011*; *Nadkarni et al., 2012*; *Scimemi and Diamond, 2012*; *Schmidt et al., 2013*; *Vyleta and Jonas, 2014*). It has been suggested that tight coupling is primarily used at synapses designed for fast, reliable transmission, whereas loose coupling is utilized at synapses specialized on presynaptic plasticity (*Bucurenciu et al., 2008*; *Eggermann et al., 2012*; *Nadkarni et al., 2012*; *Vyleta and Jonas, 2014*). The present results are fully consistent with this hypothesis, since GABAergic BC–PC synapses show fast transmitter release following single action potentials and reliable transmission during trains of spikes (*Caillard et al., 2000*; *Sakaba, 2008*; *Eggermann and Jonas, 2012*).

The present results identify novel links between the $[Ca^{2+}]_o$ independence of the TCR and nanodomain coupling. Our model reveals that the $[Ca^{2+}]_o$ dependence is substantially more prominent in loose than in tight coupling regimes in a variety of conditions. These include different $Ca^{2+}$ sensor rates, different bouton diameters, various concentrations and affinity values of endogenous buffers, and uniform vs non-uniform coupling. Specifically, our model predicts that slowing of sensor rates should markedly enhance the $[Ca^{2+}]_o$ dependence of the TCR in a microdomain, but not in a nanodomain coupling regime. We tested this prediction by lowering the temperature, which, among other potential effects, is expected to slow sensor rates. Whereas lowering the temperature had large effects on the absolute value of the half-duration of the TCR, its $[Ca^{2+}]_o$ independence was unchanged. These findings provide experimental evidence that tight source–sensor coupling is a key factor that ensures the $[Ca^{2+}]_o$ independence of the TCR.

Previous studies highlighted several functional advantages of nanodomain coupling, including efficacy, speed, temporal precision, and energy efficiency of synaptic transmission (*Bucurenciu et al., 2008*; *Eggermann et al., 2012*). Our results suggest another functional benefit: conveying $[Ca^{2+}]_o$ independence to the TCR. Is the invariance of the TCR relevant for microcircuit function under physiological conditions? Maintenance of speed and temporal precision at the GABAergic BC–PC synapse is of critical importance for the operation of the cerebellum, since feedforward inhibition mediated by BCs sets the temporal window of signal integration in PCs (*Mittmann et al., 2005*; *Bao et al., 2010*). However, $[Ca^{2+}]_o$ has been shown to fluctuate during repetitive activity, high-frequency network oscillations, and in pathophysiological conditions (*Heinemann et al., 1977*; *Borst and Sakmann, 1999*; *Rusakov and Fine, 2003*). Furthermore, $Ca^{2+}$ inflow will be changed by neuromodulators, which often act via inhibition of presynaptic $Ca^{2+}$ channels (*Takahashi et al., 1998*). Thus, nanodomain coupling at BC–PC synapses may ensure constant timing of fast feedforward inhibition under a variety of network conditions.

## Materials and methods

### Cerebellar slice preparation

Slices were cut from the cerebellum of 14- to 16-day-old C57/Bl6 wild-type mice of either sex. Experiments were performed in strict accordance with institutional, national, and European guidelines for animal experimentation. Mice were maintained under light (7 am–7 pm) and dark cycle (7 pm–7 am) conditions and were kept in a litter of 8 animals together with the mother in a single cage. Animals were lightly anesthetized using isoflurane (Forane, AbbVie, Austria) and sacrificed by rapid decapitation. The brain was rapidly dissected out and immersed in ice-cold slicing solution containing 87 mM NaCl, 25 mM NaHCO$_3$, 2.5 mM KCl, 1.25 mM NaH$_2$PO$_4$, 10 mM D-glucose, 75 mM sucrose, 0.5 mM CaCl$_2$, and 7 mM MgCl$_2$, (pH 7.4 in 95% O$_2$ / 5% CO$_2$, ~326 mOsm). Parasagittal 250-μm-thick cerebellar slices from the vermis region were cut using a custom-built vibratome. After ~20 min incubation at ~35°C, the slices were stored at room temperature. Experiments were performed at 21–23°C, unless specified differently, or at either 11–13°C or 32–35°C in subsets of experiments as indicated.

### Paired recordings

During experiments, slices were superfused with a bath solution containing 125 mM NaCl, 2.5 mM KCl, 25 mM NaHCO$_3$, 1.25 mM NaH$_2$PO$_4$, 25 mM D-glucose, 2 mM CaCl$_2$, and 1 mM MgCl$_2$ (pH 7.4 in 95% O$_2$ / 5% CO$_2$, ~316 mOsm). To investigate the relationship between IPSC peak amplitude and [Ca$^{2+}$]$_o$, different combinations of [Ca$^{2+}$]$_o$ / [Mg$^{2+}$]$_o$ were used (0.5 / 2.5, 0.6 / 2.4, 0.7 / 2.3, 0.8 / 2.2, 1 / 2, 3 / 1, 4 / 1, and 10 / 1 mM). Paired recordings from synaptically connected BCs and PCs were performed as previously described (*Caillard et al., 2000*; *Sakaba, 2008*; *Eggermann and Jonas, 2012*). Intracellular solution used for presynaptic BCs contained 125 mM K-gluconate, 20 mM KCl, 10 mM HEPES, 10 mM phosphocreatine, 2 mM MgCl$_2$, 0.1 mM EGTA, 2 mM ATP, 0.4 mM GTP (pH adjusted to 7.3 with KOH, ~310 mOsm). In a subset of experiments, 0.2% biocytin was added. For experiments using Ca$^{2+}$ chelators, 0.1 mM EGTA was replaced with different concentrations of BAPTA (1, 3, or 10 mM) or EGTA (10 or 30 mM); the concentration of K-gluconate was reduced accordingly to maintain osmolarity. Presynaptic pipettes were fabricated from borosilicate glass tubing. Presynaptic pipette resistance was 12–15 MΩ. Intracellular solution for postsynaptic PCs contained 140 mM KCl, 10 mM HEPES, 2 mM MgCl$_2$, 10 mM EGTA, 2 mM ATP, and 2 mM QX-314 (pH adjusted to 7.3 with KOH, ~310 mOsm). To achieve small postsynaptic series resistance, leaded glass (PG10165-4, WPI, Sarasota, FL) was used to fabricate large tip-sized recording pipettes. To minimize capacitance, pipettes were coated with dental wax. Postsynaptic pipette resistance was 0.8–1.5 MΩ, resulting in a mean series resistance of 3.8 ± 0.1 MΩ (range: 2.5–7.5 MΩ, 92 cells). Postsynaptic series resistance was not compensated, but continuously monitored using 5-mV test pulses. Experiments were only analyzed if changes in series resistance in the entire recording period were less than 2 MΩ. Similarly, experiments were discarded if there was a detectable rundown of IPSC peak amplitude during the control period.

For intracellular stimulation of BCs in the whole-cell configuration under current-clamp conditions, single pulses (400 pA, 4 ms at ~22°C and ~34°C; 500 pA, 5 ms at ~12°C) were injected into the BC every 4 s (~22°C and ~34°C) or 15 s (~12°C). A holding current of ~ –50 pA was applied to maintain the resting membrane potential at ~–65 mV and to avoid spontaneous action potential generation. For cell-attached stimulation under voltage-clamp conditions (*Perkins, 2006*; *Vyleta and Jonas, 2014*), the presynaptic pipette contained a K$^+$-based intracellular solution. Action potentials were evoked by brief voltage pulses (amplitude <1 V, duration 0.1–0.2 ms). Pipette holding potential was set to −60 or −80 mV to minimize the holding current and to avoid spontaneous action potential generation. Above a threshold value, action currents in BCs and IPSCs in PCs were evoked, demonstrating reliable all-or-none activation of the synapse. Experiments in which presynaptic seal resistance went below 1 GΩ during recording were discarded. In all experiments, PCs were recorded under voltage-clamp conditions at a holding potential of −70 mV. Membrane potentials given were not corrected for liquid junction potentials.

Temperature control of bath solution was achieved using a temperature controller (Sigmann, Germany) in combination with a high precision electronic thermometer (GHM, Germany) placed near the specimen. For cooling, the inflow tubing was placed in an ice reservoir.

Peptide toxins were applied using a recirculation system with a peristaltic pump (Ismatec, Germany). The total volume of the system was ~4 ml, and the solution was equilibrated with 95% O$_2$ / 5% CO$_2$. Bovine serum albumin (Sigma-Aldrich, St. Louis, MI) was added at a concentration of 1 mg ml$^{-1}$ to

eLIFE Short report

Neuroscience

prevent adsorption of peptides to the surfaces of the perfusion system. ω-agatoxin IVa and ω-conotoxin GVIa were from Bachem (Switzerland), (1,2,5,6-tetrahydropyridine-4-yl)-methylphosphinic acid (TPMPA) and CGP55845 hydrochloride were from Tocris (UK), other chemicals were obtained from Sigma–Aldrich or Merck (Germany), unless specified differently.

## Biocytin labeling

For analysis of neuron morphology, slices were fixed >24 hr in 2.5% paraformaldehyde, 1.25% glutaraldehyde, and 15% saturated picric acid in 100 mM phosphate buffer (PB; pH 7.35). After fixation, slices were washed, incubated in 2% hydrogen peroxide, and shock-frozen in liquid nitrogen. Subsequently, the tissue was treated with PB containing 1% avidin–biotinylated horseradish peroxidase complex (ABC; Vector Laboratories, Burlingame, CA) overnight at 4°C. Excess ABC was removed by several rinses with PB, before development with 0.05% 3,3'-diaminobenzidine tetrahydrochloride and 0.01% hydrogen peroxide. Subsequently, slices were rinsed in PB several times and embedded in Mowiol (Roth, Germany).

## Data acquisition and analysis

Data were acquired with a Multiclamp 700B amplifier (Molecular Devices, Sunnyvale, CA). Signals were filtered at 6 kHz (4-pole low-pass Bessel filter) and digitized at 20 or 50 kHz using a CED 1401plus interface (Cambridge Electronic Design, UK). Pulse generation and data acquisition were performed using FPulse 3.33 (U. Fröbe, Physiological Institute Freiburg, Germany) running under Igor Pro 6.3.2 (Wavemetrics, Portland, OR) on a PC. Data were analyzed with Igor Pro 6.3.2 and Mathematica 8.0.1 or 9.01 (Wolfram Research, Champaign, IL). Synaptic latency (the steepest point of the rise phase of the presynaptic action potential to the onset of the IPSC), and proportion of failures were determined from 50 to 800 single traces. To quantify the block of transmitter release by BAPTA and EGTA, the peak amplitude of the evoked IPSCs against experimental time was fit with an exponential or sigmoidal function, and the amount of suppression was quantified as the ratio of steady-state to initial values of the fit curve. Concentration–effect curves (IPSC vs $[Ca^{2+}]_o$ or [chelator]) were fit with a Hill equation of the form $f(c) = a [1 + (c_{50} / c)^n]^{-1}$, where c is concentration, a is maximal amplitude, $c_{50}$ is half-maximal effective ($EC_{50}$) or inhibitory concentration ($IC_{50}$), and n is Hill coefficient.

## Deconvolution analysis

The time course of release (TCR) was determined by deconvolution (***Diamond and Jahr, 1995***; ***Neher and Sakaba, 2001***; ***Sakaba, 2008***). Average unitary IPSCs ($IPSC_{unitary}$) were deconvolved from average quantal IPSCs ($IPSC_{quantal}$) as $F^{-1}[F(IPSC_{unitary}) / F(IPSC_{quantal})]$, where F is the discrete Fourier transform and $F^{-1}$ is the inverse. Unitary IPSCs were aligned to the steepest point in the rising phase of the corresponding presynaptic action potentials (to account for minimal jitter in the timing of action potential initiation). Quantal IPSCs recorded in conditions of reduced $[Ca^{2+}]_o$ were aligned by the rising phase of the individual events and then averaged. For subsequent analysis and display, the TCR was filtered at 5 kHz (for recordings at ~22°C or ~34°C) or 0.5 kHz (for measurements at ~12°C) using a digital filter. Finally, the filtered TCR was fit with a Gaussian function. The effects of filtering were corrected by subtracting the variance of an impulse response of a Gaussian filter (***Colquhoun and Sigworth, 1995***). At 2 mM $[Ca^{2+}]_o$, the quantal content, estimated as the integral under the TCR, was 10.4 ± 1.8. Analysis was performed using Mathematica 8.0.1 running under Windows 7 on a PC.

To test the possibility that postsynaptic factors, such as receptor desensitization, receptor saturation, or GABA spillover affected our measurements, we measured the TCR in the presence of the low-affinity competitive antagonist 300 μM TPMPA (***Jones et al., 2001***) and 2 μM CGP55845 to avoid effects of TPMPA on $GABA_B$ receptors. In 4 mM $[Ca^{2+}]_o$ (presumably maximizing the confounding effects of postsynaptic factors), the half-duration of the TCR was 0.51 ± 0.04 in the presence (***Figure 2—figure supplement 2***; 6 pairs) vs 0.47 ± 0.02 in the absence of TPMPA (***Figure 2E***; p = 0.38).

## Estimation of distance between Ca²⁺ channels and release sensors

To quantify the coupling distance between $Ca^{2+}$ channels and release sensors, concentration–effect data for both BAPTA and EGTA were first fit with a Hill equation. Next, data were analyzed with a model of $Ca^{2+}$ diffusion and buffering based on linear approximations (***Neher, 1998***; ***Bucurenciu et al., 2008***). The ratio of $Ca^{2+}$ transients in the presence and absence of chelators was converted into the ratio of release probabilities, using a power function with the Hill coefficient set according to the

slope of the double-logarithmic IPSC–$[Ca^{2+}]_o$ relation in the low-concentration limit (**Figure 2B**, bottom). Three different model variants were used: (1) a model with a constant coupling distance, (2) a model with a half-normally distributed coupling distance, and (3) a model with skewed-normally distributed coupling distance. In the two latter cases, the average coupling distance was specified as the expectation value of the distribution. For BAPTA, the $Ca^{2+}$ binding and unbinding rates were assumed as $k_{on} = 4\ 10^8\ M^{-1}s^{-1}$ and $k_{off} = 88\ s^{-1}$ (affinity 220 nM). For EGTA, the rates were taken as $k_{on} = 1\ 10^7\ M^{-1}s^{-1}$ and $k_{off} = 0.7\ s^{-1}$ (affinity 70 nM). The diffusion coefficients for $Ca^{2+}$, EGTA, and BAPTA were assumed to be 220 $\mu m^2\ s^{-1}$ (**Neher, 1998**). The endogenous buffer product was set to 5500 $s^{-1}$ and the resting $Ca^{2+}$ concentration was assumed as 40 nM (**Collin et al., 2005**). Confidence intervals of coupling distance were obtained by bootstrap procedures. 1000 artificial data sets were generated from the means and SEMs of the original data set and analyzed as the original (**Efron and Tibshirani, 1998**). Error estimates were given as half of the 15.9–84.1 percentile range. All simulations were performed using Mathematica 8.0.1 running under Windows 7 on a PC.

## Statistics

All values are given as mean ± SEM. Error bars in the figures also indicate SEM (shown only if larger than symbol size). Statistical significance was tested using a two-sided Wilcoxon signed rank test for paired data, a two-sided Wilcoxon rank sum test for unpaired data, and a Kruskal–Wallis test for multiple comparisons (Igor Pro 6.3.2). Differences with $p < 0.05$ were considered significant.

## Modeling of transmitter release

$Ca^{2+}$ diffusion and binding to mobile and fixed buffers were modeled using the full set of partial differential equations (PDEs) of the reaction–diffusion problem, including all necessary boundary and initial conditions (**Smith, 2001**; **Bucurenciu et al., 2008**; **Eggermann and Jonas, 2012**; **Vyleta and Jonas, 2014**). PDEs were solved numerically with NDSolve of Mathematica 8.01 running under Windows 7 on a PC. A release unit was implemented as a hemisphere (**Figure 3A**). The diameter was assumed to be 1 μm, approximately corresponding to the radius of inhibitory boutons, unless specified differently. PDEs were integrated over the radial coordinate (2000 grid points) and solved at a concentration accuracy of 0.01 nM.

Brief single action potentials were applied as stimuli. A cluster of $Ca^{2+}$ channels was represented as a point source. A previously published Hodgkin-Huxley-type gating model of P/Q-type $Ca^{2+}$ channels was used to calculate the $Ca^{2+}$ inflow (**Borst and Sakmann, 1998**). The $Ca^{2+}$ inflow, relative to that of a single $Ca^{2+}$ channel, was varied between 1 and 100. The single-channel conductance was assumed as 2.2 pS (**Li et al., 2007**). The coupling distance was varied between 10 and 200 nm.

The standard parameters of the model were as follows: For the fixed endogenous $Ca^{2+}$ buffer, the rates were chosen as $k_{on} = 5\ 10^8\ M^{-1}\ s^{-1}$ and $k_{off} = 1000\ s^{-1}$ (affinity 2 μM). For standard simulations, both a fixed buffer (100 μM) and a mobile buffer with BAPTA-like properties (10 μM) were incorporated. $Ca^{2+}$ buffer concentrations were considered to be spatially uniform. The resting $Ca^{2+}$ concentration was set to 40 nM (**Collin et al., 2005**). Vesicular release rate was computed using a model of transmitter release originally established at the calyx of Held (**Lou et al., 2005**). This model was preferred over alternative models because it is based on the most extensive set of experimental data. The occupancies for the different states of the model were obtained by solving the corresponding first-order ordinary differential equations with a Q-matrix approach. Release rate was computed as the sum of the product of occupancy and release rate for each state; pool depletion was not considered.

To account for the rapid timing of transmitter release at BC–PC synapses, the previously used presynaptic action potential (**Meinrenken et al., 2002**; **Bucurenciu et al., 2008**) was time-compressed by a factor of two, $Ca^{2+}$ channel gating rates were multiplied by a factor of two, and the binding and unbinding rates of the sensor were increased by a factor of two. The maximal release rate in the model was 6008 $s^{-1}$ (**Lou et al., 2005**). In a subset of simulations, a distributed arrangement of release sensors was assumed, with a coefficient of variation for the coupling distance of 0.3 (**Figure 3E**). In these cases, release rates were obtained at various distances, and the average release rate was computed as the weighted mean, with weight factors set according to a normal distribution.

## Acknowledgements

We thank Drs Nicholas Vyleta and Ryuichi Shigemoto for critically reading the manuscript. We also thank F Marr and M Duggan for technical assistance, and A Solymosi for manuscript editing.

# Additional information

## Funding

| Funder | Grant reference number | Author |
|---|---|---|
| Austrian Science Fund | P 24909-B24 | Peter Jonas |
| European Research Council | Advanced Grant 268548 | Peter Jonas |

The funders had no role in study design, data collection and interpretation, or the decision to submit the work for publication.

## Author contributions

IA, Acquisition of data, Analysis and interpretation of data, Drafting or revising the article; PJ, Conception and design, Analysis and interpretation of data, Drafting or revising the article

## Ethics

Animal experimentation: Experiments were performed in strict accordance with institutional, national, and European guidelines for animal experimentation and were approved by the Bundesministerium für Wissenschaft, Forschung und Wirtschaft (A Haslinger, Vienna). Mice were maintained under light (7 am–7 pm) and dark cycle (7 pm–7 am) conditions and were kept in a litter of eight animals together with the mother in a single cage. Animals were lightly anesthetized using isoflurane (Forane, AbbVie, Austria) and sacrificed by rapid decapitation.

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
