## [Decision Letter]

Thank you for sending your work entitled “Nanodomain coupling explains Ca^2+^ independence of transmitter release time course at a fast central synapse” for consideration at *eLife.* Your article has been favorably evaluated by Eve Marder (Senior editor), Michael Häusser (Reviewing editor), and 2 reviewers.

The Reviewing editor and the other reviewers discussed their comments before we reached this decision, and the Reviewing editor has assembled the following comments to help you prepare a revised submission.

This manuscript examines an interesting and under-studied question, namely whether synaptic delays vary with external calcium concentrations and if not, why. To address this, the authors have performed paired recordings at basket cell Purkinje cell synapses, in combination with computational modelling. They include a study of latencies at two temperatures, a pharmacological analysis of synaptic calcium channel subtypes, a deconvolution-based analysis of release rate as a function of external calcium, and an investigation of the effects of added EGTA and BAPTA on synaptic efficacy. The results are then simulated using a variant of the previously published Eggermann-Jonas model at the same synapse.

The experimental results, which are of high quality, clearly show that there is tight coupling between SVs and VDCCs and also, that the release rate does not vary significantly with external calcium at physiological temperature. While intuition suggests that a causal link between these two findings is likely, the simulation of Figure 3 brings disappointingly little to support this hypothesis, or to eliminate alternative explanations. As detailed below, the possibility remains open that other aspects of the synapse function such as width of the AP, speed of the release reactions, or effects of endogenous buffers, may play a key role as well. Repeating the experiments of Figure 2 at room temperature should help to solve this problem. Therefore, we believe these experiments are essential for the resubmission, should not be very time-consuming, but will be very informative.

Further detail is provided below:

1) The figure describing the simulations (Figure 3) is difficult to understand in detail. The main result is in panel c. But there seems little justification in using the slope ratio at 200 s-1 to evaluate the effect of coupling distance on calcium sensitivity. It would be better to use the slope calculated by joining together the points representing 1, 2 and 4 mM calcium (respectively near 200, 2000 and 4000 s-1). If this is done it appears that the changes in TCR duration are only near 15 or 20% for the loose coupling (roughly decreasing from 0.65 to 0.55). A similar analysis made on panel e (including a dispersion in coupling distances) leads to even smaller changes. Looking at these simulations, it does seem like the dependence on external calcium is less for tight coupling than for loose coupling. But in either case the dependence is modest, presumably because the model has been constrained to mimic a very fast synapse (fast AP, fast calcium channel, fast SV release). So at the end it is not clear whether the weak dependence of TCR on external calcium should be ascribed primarily to the tight coupling as proposed by the authors, or to other features of the model.

2) Additional room temperature experiments would significantly strengthen the manuscript and should be carried out prior to resubmission. The results of Figure 1 show a much broader distribution of latencies at 22^°^C than at 34^°^C. Likewise the previous deconvolution analysis by Sakaba at the same synapse (Neuron 57, 406, 2008), which was performed at room temperature, indicates a broader TCR than in Figure 2 here. This suggests an opportunity to test whether it is really the tight coupling that prevents latency changes with extracellular calcium, or whether other factors play a role as well. If in spite of the broader TCR, external calcium effects remain negligible at room temperature, then the case of tight coupling as a causal link will be much strengthened. If however significant changes occur at room temperature, then additional synaptic parameters will have to be considered as well.

3) The SV release rates appear to be per synapse in Figure 2, and per vesicle in Figure 3, making the correspondence between the two figures difficult. The “default” rate in Figure 3 is 2000 s-1 presumably corresponding to the 15 quanta /ms of Figure 2 at 2mM calcium. How the 2000 s-1 rate was obtained is unclear. It may have been obtained by extrapolating the normalized IPSC data of Figure 2 to very high calcium, and inferring that the release rate at 2mM is 1/3 of the maximum, that is a third of 6000 s-1, or 2 000 s-1. However the extrapolation cannot be considered as accurate because high calcium points are difficult to obtain at physiological temperatures, so that the asymptote is not well constrained in Figure 2. If true, the 2 000 s-1 value would indicate an RRP value of 15/2=7.5 per synapse, much smaller than the mean value of 110 found by [33]. Overall vesicular release rates may have been seriously overestimated in the simulation of Figure 3.

4) This paper comes in a line of other works by the senior author and others measuring the channel sensor coupling in different synapses. A common conclusion is that at GABAergic synapses the coupling is 'tight'. In this context, some of the results presented here are not particularly surprising (e.g. all of Figure 1, parts of Figure 2, and all of Figure 4). It would be helpful if the authors could emphasize more strongly which of the results they consider to be particularly novel and important, especially in light of the recent literature (see next comment).

5) The discussion of the literature needs to be expanded. While it is true that latency variations have been little investigated in the CNS, the literature is not as empty as the manuscript would suggest. For example, Boudkkazi et al (Neuron 56, 1048, 2007) found that delays vary with release probability at cortical synapses, including when increasing the extracellular calcium concentration. Gilmanov IR et al 2008 also approached these issues at NMJ with similar conclusions. Furthermore, the Ca independence of release has long been noted at NMJ (as the authors acknowledge), but the authors don't refer to any studies of this in neurons, though there are several which could be cited and discussed, starting perhaps with Silver et al. 1996 in granule cells.

---

## [Author Response]

*1) The figure describing the simulations (*Figure 3*) is difficult to understand in detail. The main result is in panel c. But there seems little justification in using the slope ratio at 200 s-1 to evaluate the effect of coupling distance on calcium sensitivity. It would be better to use the slope calculated by joining together the points representing 1, 2 and 4 mM calcium (respectively near 200, 2000 and 4000 s-1). If this is done it appears that the changes in TCR duration are only near 15 or 20% for the loose coupling (roughly decreasing from 0.65 to 0.55). A similar analysis made on panel e (including a dispersion in coupling distances) leads to even smaller changes. Looking at these simulations, it does seem like the dependence on external calcium is less for tight coupling than for loose coupling. But in either case the dependence is modest, presumably because the model has been constrained to mimic a very fast synapse (fast AP, fast calcium channel, fast SV release). So at the end it is not clear whether the weak dependence of TCR on external calcium should be ascribed primarily to the tight coupling as proposed by the authors, or to other features of the model*.

We have used the concentration–effect curve in Figure 2 to estimate the release rates for our experimental conditions (0.7, 1, 2, and 4 mM [Ca^2+^]_o_; we have included additional experiments to determine curve parameters more precisely, see point 3 below). We have estimated that the peak release rates at 0.7 and 1 mM [Ca^2+^]_o_ are 168 and 379 s^-1^, respectively. We have therefore decided to maintain our original reference point (200 s^-1^), which is exactly in-between. Finally, we have indicated our experimental release rates with red arrows in the graphs of Figure 3. These data corroborate that if there were marked dependencies of TCR on release rate, as e.g. the case in a loose coupling regime, we would be able to detect this dependence in our experiments. We thank the editor / reviewers for raising this point.

*2) Additional room temperature experiments would significantly strengthen the manuscript and should be carried out prior to resubmission. The results of*
Figure 1
*show a much broader distribution of latencies at 22*^*°*^*C than at 34*^*°*^*C. Likewise the previous deconvolution analysis by Sakaba at the same synapse (Neuron 57, 406, 2008), which was performed at room temperature, indicates a broader TCR than in*
Figure 2
*here. This suggests an opportunity to test whether it is really the tight coupling that prevents latency changes with extracellular calcium, or whether other factors play a role as well. If in spite of the broader TCR, external calcium effects remain negligible at room temperature, then the case of tight coupling as a causal link will be much strengthened. If however significant changes occur at room temperature, then additional synaptic parameters will have to be considered as well*.

We thank the editors / reviewers for this excellent suggestion. We would like to point out that the results shown in the original submission (with the exception of Figure 1) had already been performed at room temperature. We have revised the Materials and methods section and the Figure legends to make this point more clear. We are aware that the TCR in our experiments (0.47 ± 0.01 ms) is shorter than the approximate value reported by [33] (∼1 ms) under seemingly identical conditions. However, because of the high Q_10_ value of the TCR (see below), this may have been caused by relatively minor differences in temperature.

To constructively pick up the key idea of the editors / reviewers, we performed additional experiments at lower temperature (∼12°C; 5 pairs). Under these experimental conditions, kinetic parameters (presynaptic action potential duration, presynaptic Ca^2+^ channel gating, and release sensor kinetics) are expected to be slowed down, whereas tight coupling is likely to be maintained. These experiments turned out to be surprisingly difficult, because of lower stimulation frequencies required to ensure pool replenishment and several additional experimental issues related to low temperature. Interestingly, reduction in temperature markedly prolonged the TCR, whereas the dependence of TCR on [Ca^2+^]_o_ remained minimal. These results (incorporated into Figure 2—figure supplement 1) corroborate our original conclusion that the tight coupling is the most important parameter in shaping the Ca^2+^-independence of transmitter release. Again, we thank the editor / reviewers for this suggestion.

*3) The SV release rates appear to be per synapse in*
Figure 2*, and per vesicle in*
Figure 3*, making the correspondence between the two figures difficult. The “default” rate in*
Figure 3
*is 2000 s-1 presumably corresponding to the 15 quanta /ms of*
Figure 2
*at 2mM calcium. How the 2000 s-1 rate was obtained is unclear. It may have been obtained by extrapolating the normalized IPSC data of*
Figure 2
*to very high calcium, and inferring that the release rate at 2mM is 1/3 of the maximum, that is a third of 6000 s-1, or 2 000 s-1. However the extrapolation cannot be considered as accurate because high calcium points are difficult to obtain at physiological temperatures, so that the asymptote is not well constrained in*
Figure 2*. If true, the 2 000 s-1 value would indicate an RRP value of 15/2=7.5 per synapse, much smaller than the mean value of 110 found by*
[33]*. Overall vesicular release rates may have been seriously overestimated in the simulation of*
Figure 3.

We have performed additional experiments to better constrain the concentration–effect curve in the saturation phase, as requested by the reviewer (5 pairs at 10 mM [Ca^2+^]_o_). The results are now included in Figure 2, and better constrain our estimates of release probability. The new results confirm that the “physiological” release probability at 2 mM [Ca^2+^]_o_ is 0.26, and that the corresponding peak release rate is close to 2000 s^-1^. Given that we extended the range of [Ca^2+^]_o_, we further decided to determine the Hill coefficient in the concentration range ≤ 2 mM rather than the entire concentration range. We have included a double-logarithmic representation to graphically illustrate this point (Figure 2, bottom of the revised version). Please note that this different (presumably more accurate) measurement of the Hill coefficient also slightly changed the results of our analysis of coupling distance (in the main text and legend of Figure 4).

We also thank the reviewers for reminding us of the paper by [33]. We have cited this paper in the revised version. We are aware that our mean peak release rate and the relatively high release probability may suggest that the size of the releasable pool is small. Our simple quantitative estimation is pool size = quantal content (from deconvolution and integral of TCR) / release probability (from Ca^2+^ dependence of release) = 10.4 / 0.26 = 40 at 2 mM [Ca^2+^]_o_. In apparent contrast, the cumulative release analysis during trains of action potentials and Ca^2+^ uncaging suggested a larger pool size (110 ± 28; [33]). One possible explanation might be that the size of the pool accessed by single action potentials is smaller than that accessed by trains or Ca^2+^ uncaging. Furthermore, there are many additional complications in the analysis of train data, including release-independent components of depression (Kraushaar and Jonas, 2000), Ca^2+^- and time-dependent changes in refilling rate (Wang and Kaczmarek, 1998, Nature), and nonlinearities in the extrapolation in cumulative release plots (Thanawala and Regehr, 2013, J. Neuroscience). Thus, whether or not there is a true discrepancy remains to be resolved.

*4) This paper comes in a line of other works by the senior author and others measuring the channel sensor coupling in different synapses. A common conclusion is that at GABAergic synapses the coupling is 'tight'. In this context, some of the results presented here are not particularly surprising (e.g. all of*
Figure 1*, parts of*
Figure 2*, and all of*
Figure 4*). It would be helpful if the authors could emphasize more strongly which of the results they consider to be particularly novel and important, especially in light of the recent literature (see next comment)*.

We have completely changed the Discussion to indicate the novel aspects more clearly. We continue to think that the manuscript provides interesting data, as also stated by the editor’s summary that the paper addresses an “interesting and under-studied question”. Furthermore, not all papers converge on the conclusion that coupling is tight. For example, we have recently shown loose coupling in hippocampal mossy fiber synapses (41), and earlier papers suggested loose coupling at CA3–CA1 cell synapses based on the properties of synaptic dynamics (26). Likewise, previous studies using membrane-permeant Ca^2+^ chelators indicated loose coupling in CCK interneuron output synapses (18). Finally, the suggestions of the editor / reviewers have helped to convey an additional novel aspect to the paper, because, to the best of our knowledge, nobody ever studied the [Ca^2+^]_o_ dependence of the TCR at different temperatures.

*5) The discussion of the literature needs to be expanded. While it is true that latency variations have been little investigated in the CNS, the literature is not as empty as the manuscript would suggest. For example, Boudkkazi et al (Neuron 56, 1048, 2007) found that delays vary with release probability at cortical synapses, including when increasing the extracellular calcium concentration. Gilmanov IR et al 2008 also approached these issues at NMJ with similar conclusions. Furthermore, the Ca independence of release has long been noted at NMJ (as the authors acknowledge), but the authors don't refer to any studies of this in neurons, though there are several which could be cited and discussed, starting perhaps with Silver et al. 1996 in granule cells*.

We thank the reviewers for reminding us of the papers of [17], Silver et al., 1996 (and [34]). Although the TCR is not directly reported in those studies, we agree that these papers are relevant to our work, because they report Ca^2+^-dependence of the EPSC time course. We have incorporated [17] and [34], as suggested. However, we would like to add that latency and time course of release should not be confused. For example, Boudkkazi et al. studied latency variation rather than TCR (thus, those results will not be directly relevant to the present paper).